

# A 5-channel cavity ring-down spectrometer for the detection of NO₂, NO₃, N₂O₅, total peroxy nitrates and total alkyl nitrates

N. Sobanski[1], J. Schuladen[1], G. Schuster[1], J. Lelieveld[1] and J. N. Crowley[1]

[1]Atmospheric Chemistry Department, Max-Planck-Institut für Chemie, 55128 Mainz, Germany.

*Correspondence to*: John N. Crowley (john.crowley@mpic.de)

**Abstract.** We report the characteristics and performance of a newly developed 5-channel cavity-ring-down spectrometer to detect $NO_3$, $N_2O_5$, $NO_2$, total peroxy nitrates ($\Sigma$PNs) and total alkyl nitrates ($\Sigma$ANs). $NO_3$ and $NO_2$ are detected directly at 662 nm and 405 nm, respectively. $N_2O_5$ is measured as $NO_3$ after thermal decomposition at 383 K. PNs and ANs are detected as $NO_2$ after thermal decomposition at 448 K and 648 K. We describe details of the instrument construction and operation as well as the results of extensive laboratory experiments that quantify the chemical and optical interferences that lead to biases in the measured mixing ratios, in particular involving the reactions of organic radical fragments following thermal dissociation of PNs and ANs. Finally, we present data obtained during the first field deployment of the instrument in July 2015.

## 1 Introduction

Nitrogen oxides play a central role in tropospheric chemistry though their influence on ozone production and radical cycling, and thus on the oxidative capacity of the atmosphere. Nitric oxide (NO) is directly emitted into the boundary layer by e.g. fossil fuel combustion and biomass burning, and is oxidised via various mechanisms (e.g. reaction with $O_3$ and peroxy radicals, $RO_2$) to form $NO_2$, which is the dominant precursor of photochemically formed, tropospheric ozone. Together, NO and $NO_2$ are referred to as NO*x*, while the sum of all reactive nitrogen species (NO*x* plus inorganic and organic nitrates and nitrites and halogenated nitrogen oxides) is referred to as NO*y*. In the tropospheric boundary layer, $NO_2$ mixing ratios vary from <10 pptv in remote regions to >100 ppbv in highly polluted environments. Tropospheric $NO_2$ has been measured with several techniques including Differential Optical Absorption Spectroscopy (DOAS), Chemiluminescence (CLD), Laser Induced Fluorescence (LIF) and Cavity-Ring-Down Spectroscopy (CRDS). For a summary and inter-comparison of these methods, the reader is referred to Fuchs et al. (2010). $NO_2$ is the major precursor of the inorganic nitrates $NO_3$ and $N_2O_5$. During day-time, $NO_3$ is rapidly photolysed (Johnston et al., 1996) and also reacts with NO to reform $NO_2$ so that $NO_3$ mixing ratios are very low. At night, $NO_3$ levels may reach 100s of pptv in polluted air and their reactions represent a significant route for oxidation of VOCs and, via formation of $N_2O_5$, are a source of particulate nitrate and thus sink of $NO_x$ (Wayne et al., 1991; Allan et al., 1999). Tropospheric $NO_3$ has been detected using long-path, LP-DOAS (Platt et al., 1980)



over a path length of typically > 1 km, whilst point detection methods of $NO_3$ include cavity enhanced DOAS, LIF and CRDS based instruments. $N_2O_5$ has been detected as $NO_3$ after thermal decomposition or directly measured by chemical ionization mass spectrometry. Descriptions of methods for detecting $NO_3$ and $N_2O_5$ and an inter-comparison experiment have recently been published (Fuchs et al., 2012; Dorn et al., 2013).

Reactions between atmospheric oxidants such as OH or $NO_3$ and anthropogenic or biogenic volatile organic compounds (VOCs) lead to the formation of organic peroxy radicals that react with NO or $NO_2$ to form organic nitrates (Roberts, 1990a). Reaction of $RO_2$ with $NO_2$ results in the formation of peroxy nitrates ($RO_2NO_2$, henceforth abbreviated as PNs). If the initial peroxy radical contains a terminal acyl group ($R(O)O_2$), peroxyacetic nitric anhydrides (also called peroxyacyl nitrates or PANs, $R(O)O_2NO_2$) are formed, which have lifetimes on the order of hours in the temperate boundary layer conditions but

weeks in cold regions of the atmosphere. Following generation and transport away from pollution sources, PANs can thermally decompose to $NO_2$ and an organic radical, thus providing NO$x$ and enhancing the production of $O_3$ in remote locations. Non-substituted peroxy radicals react to form PNs that are less thermally stable, which undergo rapid (timescale of seconds at 298 K) thermal decomposition in the mid-latitude boundary layer and thus reach higher concentrations only in colder regions such as the upper troposphere and close to the poles (Slusher et al., 2002; Murphy et al., 2004). The reaction

of $RO_2$ with NO mainly forms RO + $NO_2$ but can also result in the formation of alkyl nitrates (RONO2, henceforth abbreviated as ANs) in a minor branch (1-30 %) of this reaction, with higher yields for long chain peroxy-radicals (Arey et al., 2001; Perring et al., 2013). ANs are also formed at night from the reaction between $NO_3$ and VOCs with a higher yield (20 to 80%) than the OH initiated process (Perring et al., 2013). The oxidation of alkenes by $NO_3$ results in the formation of nitroxy-alkyl radicals that can react with $O_2$ to form nitroxy alkyl peroxy radicals that can undergo further chemistry. ANs

are longer lived than PANs against thermal decomposition and can thus also act as NO$x$ reservoirs. Reactions with OH, photolysis or transfer to the aerosol phase are the major sinks of ANs depending on carbon chain size and degree of substitution. PNs and ANs represent a very wide group of species present in the atmosphere, with different levels of structural complexity due to the different possible combinations of functional groups. Individual nitrates can be measured using chromatography based instruments followed by electron capture detection or conversion to NO or $NO_2$ (Roberts et al.,

1989; Roberts, 1990a; Flocke et al., 1991; Blanchard et al., 1993) whereas the detection of the total (i.e. non-speciated) ambient PNs and ANs has been demonstrated using thermal decomposition of the nitrates followed by $NO_2$ detection (Day et al., 2002; Day et al., 2003; Paul et al., 2009; Paul and Osthoff, 2010; Wooldridge et al., 2010; Thieser et al., 2016).

A single instrument that measures $NO_2$, $NO_3$, $N_2O_5$ ΣPNs and ΣANs can clearly provide great insight into the reactive nitrogen budget of the troposphere, enabling studies of day- and night-time processes that transfer NO$x$ from the gas to the

particle phase or to organic reservoir species. The interconversions and relation to NO$x$ losses and photochemical $O_3$ generation of these traces-gases are described in Fig.1, and those highlighted in red circles are detected with this instrument. The 5-channel instrument we describe is a further development and extension of the two-channel, one colour (405 nm) prototype recently described by Thieser et al. (2016) and of the two-channel set-up for measuring $NO_3$ and $N_2O_5$ previously developed and operated by this group (Schuster et al., 2009). Whilst some basic concepts are similar, several significant



design modifications (optical, mechanical, electrical/data-acquisition and chemical) have been made that improve both the precision, limit of detection and accuracy of the measurements. These improvements are described in the relevant sections below.

In Sect. 2 we describe the overall design of the instrument and the features specific to the measurement of the individual trace-gases. In Sect. 3 calibration procedures and laboratory characterisation experiments are described along with an assessment of the overall uncertainty. Finally, in Sect. 4 we show data from a first field deployment at a semi-rural, mountain-site in Germany.

## 2 Instrument design

The instrument we describe uses Cavity-Ring-Down Spectroscopy (CRDS), partly coupled with thermal dissociation (TD-CRDS). CRDS is a detection technique that was developed in the early 90s (Okeefe and Deacon, 1988; Lehmann and Romanini, 1996) and has been recently reviewed (Berden et al., 2000; Brown, 2003). The underlying principle of CRDS is measurement of optical extinction in a (usually) closed cavity. In contrast to differential absorption spectroscopy in which optical extinction is linked to a measured light intensity difference, in CRDS setups the optical extinction is calculated from the rate of decay of light intensity (ring-down constant) measured at the output of the cavity after a light pulse (pulsed CRDS) or after a continuous light source is turned off (cw-CRDS). The light intensity decay, measured behind a cavity mirror after switching the light source off is exponential and the concentration of an absorbing / scattering gas can be calculated using Eq. (1)

$$[X] = \frac{1}{c\sigma_{(i,\lambda)}}\left(\frac{1}{\tau(s)} - \frac{1}{\tau_0}\right) \qquad \text{Eq. (1)}$$

Where $\tau_0$ and $\tau$ (s) correspond to decay constants in the absence and presence of absorbing / scattering trace gases, respectively (see Sect. 2.1 and 2.2). $\sigma_{(i,\lambda)}$ is the absorption cross-section / scattering coefficient of species $i$ at wavelength $\lambda$. The determination of the decay constants is done using a Linear Regression of Sums fitting method (LRS) which has a number of advantages compared to least-squares (LS) fitting (Everest and Atkinson, 2008) including a faster algorithm and removing the need to measure an accurate zero (or baseline), which enables use of shorter on-off-modulation cycles and higher data acquisition frequencies. Comprehensive comparison between the LS and LRS methods for both synthetic and real ring-down-decays showed excellent agreement.

The central part of the instrument described here consists of 5 cavities of similar design and materials (see Fig. 2). Two cavities are used to detect $NO_3$ and $N_2O_5$ (at 662 nm) and three are used to detect $NO_2$, $\Sigma$PNs and $\Sigma$ANs (at 405 nm). We refer to the 5 different inlet / cavity combinations (downstream of the filters, see Fig. 2) as the $NO_2$, $NO_3$, $N_2O_5$, $\Sigma$PNs and $\Sigma$ANs channels. The light sources for the 5 cavities are two laser diodes housed in Thorlabs TCLDM9 modules located in



aluminium housings with optical components for beam-splitting, collimation and optical isolation (see Sect. 2.1 and 2.2). Coupling between the laser diodes and the cavities is achieved using 50 μm core optical fibres with collimators adjusted to weakly focus the beam (diameter of ≈ 3 mm) at a point ≈ 1 m behind the cavity output mirrors. The cavities are constructed from 70 cm long, seamless, stainless-steel pipes (1 mm thick, with an inner diameters of 8 mm) which is Teflon coated

(DuPont, FEP, TE 9568). The use of stainless-steel as cavity material is considered superior to glass as used in our previous setups (Schuster et al., 2009; Thieser et al., 2016) as it eliminates the risk of breakage, and provides more homogeneous heating of the cavities. Radical losses on FEP-coated stainless-steel or glass surfaces are indistinguishable. PFA T-pieces (Swagelok) are mounted at both ends of the piping for admitting and exhausting the sample flow. Metal bellows / adjustable mirror supports are positioned behind each T-piece, resulting in a distance between mirrors of 93 cm. The bellows provide a

purge-gas volume for keeping the mirrors clean and reduce physical stress on the mirror supports. The 10 cavity-mirrors are continuously purged with dry, synthetic air to prevent degradation in reflectivity. Purging the mirrors has the drawback of reducing the effective cavity length and thus sensitivity but enabled uninterrupted operation over periods of several months. The 10 different purge gas streams are controlled by 2 mass flow controllers coupled with two sets of critical orifices designed for different purge flow rates depending on the size and direction of flow of ambient air through the cavities. The

purge gas flows for the 405 nm cavities are 500 cm$^3$ (STP) min$^{-1}$ (sccm) for the mirror opposing the main gas flow (downstream) and 100 sccm for the upstream mirror. For the 662 nm cavities, these flows are 350 (downstream) and 150 sccm (upstream).The light intensity exiting each cavity is measured and converted to an analogue electric signal by a photomultiplier / preamplifier device (Hamamatsu H10492-012 at 662 nm and H10492-002 at 405 nm). Temperature, pressure and flows are measured/regulated by a custom designed control unit (referred to as V25). Data acquisition (ring-

downs and laser spectra) and data processing are performed by an embedded computer (NI PXIe-8135). Analogue-to-digital conversion of the PMT signals is performed via two acquisition cards (NI PXI-6132, 14 bit, 3 MS s$^{-1}$) in the embedded computer rack, one for the two 662 nm cavities and one for the three 405 nm cavities.

**2.1 Detection of NO$_3$ and N$_2$O$_5$ at 662 nm**

NO$_3$ and N$_2$O$_5$ are detected using the strong absorption feature of NO$_3$ at about 662 nm ($\sigma_{662nm} \approx 2.3 \times 10^{-17}$ cm$^2$ molecule$^{-1}$ at

298 K (Orphal et al., 2003; Osthoff et al., 2007)). NO$_3$ is measured in one channel (the NO$_3$ channel) at room temperature. In a second channel (the N$_2$O$_5$ channel), the sum of ambient NO$_3$ and NO$_3$ arising from the thermal decomposition of N$_2$O$_5$ at 383 K are measured, enabling the N$_2$O$_5$ mixing ratio to be calculated from the difference. The cavity mirrors (Advanced Thin Films, 1" diameter, 1 m radius of curvature) have a nominal reflectivity of 0.999985 and result in ring-down times (at atmospheric pressure of dry synthetic air) of 150 - 160 μs. To keep a constant emission wavelength of ≈ 662 nm, the laser

diode (Thorlabs HL6545MG) is thermostatted to 38 °C and has an emission bandwidth of ≈ 0.5 nm (fwhm). An optical isolator is positioned at the immediate output of the diode laser to prevent back reflexions and the final intensity of light reaching the front mirror of each of the two red cavities is ≈ 10 to 15 mW. The laser wavelength is monitored using a mini spectrograph (Ocean Optics, type HR4C2509) which records diffuse back reflexions from the N$_2$O$_5$ cavity input mirror.





Taking into account the emission spectrum and the absorption cross-section of $NO_3$, the effective cross-section obtained during the first field deployment and calibrations is $\approx 2.1 \times 10^{-17}$ cm$^2$ molec$^{-1}$. The uncertainty on the effective cross-section is discussed in Sect. 3.1.

Before entering the $NO_3$ and $N_2O_5$ channels air ($\approx 15$ L (STD) min$^{-1}$, slm) is drawn through a filter (Pall Corp., Teflon membrane, 47 mm Ø, 0.2 µm pore ) housed in an automatic filter changer. The changer has a maximum capacity of 20 to 25 filters, allowing for 2-3 nights of measurement (at a change rate of one filter per hour) without requiring the presence of an operator. Losses of $NO_3$ in the filter and filter-changer are discussed in Sect. 3.1. Air exits the filter changer through a ¼ PFA pipe ($\approx 30$ cm) with a T-piece to enable the injection of a few sccm of NO (100 ppmv in $N_2$) for titrating $NO_3$ and zeroing the instrument, which was performed for a few seconds every four minutes. The air then passes through another T-piece that divides the flow between the $NO_3$ (8 slm) and $N_2O_5$ (7 slm) channels. For $N_2O_5$ detection, the air flows through a heated, Teflon-coated heated glass section (wall temperature 383 K) to decompose $N_2O_5$ into $NO_3 + NO_2$. This temperature is sufficient to decompose all the $N_2O_5$, which was verified experimentally by increasing the temperature until the maximum signal was obtained. The residence time in this reactor ($\approx 0.2$ s) is sufficient to allow for a reaction between total $NO_3$ and NO to go to completion during zeroing periods. In the $NO_3$ channel, the air flows first though FEP-coated glass tubing at room temperature to ensure sufficient time for complete $NO_3$ titration when adding NO (i.e. when zeroing). Residence times are 0.45 s in the $NO_3$ channel and 0.38 s in the $N_2O_5$ channel. The pressure in the $NO_3$ and the $N_2O_5$ channels is not regulated and depends on atmospheric pressure and flow rates in the instrument.

A square-wave signal provided by the V25 is used to modulate the diode laser on and off and also triggers the acquisition of the individual decays. Typically, the modulation frequency is 400 Hz, with on and off times of 1 and 1.5 ms. Averaging 400 decay constants thus resulting in an instrumental time resolution of 1 data-point per second. An Allan deviation plot obtained under laboratory conditions (800 mbar cavity pressure) for the $NO_3$ and $N_2O_5$ cavities is shown in Fig. 3. The Allan deviation plot shows that the ($1\sigma$) standard deviations of the zero signal for the $NO_3$ and $N_2O_5$ channels for a 1 s integration time are 0.1 and 2 pptv, respectively. This is found to be a significant improvement in signal-to-noise ratio compared to the instrument described by Schuster et al. (2009), for which 5 s integration times were necessary to reduce noise levels to $\approx 2$ pptv for both $NO_3$ and $N_2O_5$ channels. The improvement is likely related to the more stable optical set-up. Despite the fact that the two 662 nm channels in the new instrument are mechanically and optically very similar, the $N_2O_5$ channel shows a higher noise level, which is related to enhanced turbulence due to temperature and thus density gradients (Osthoff et al., 2006).

## 2.2 Detection of $NO_2$, $\Sigma PNs$ and $\Sigma ANs$ at 405 nm

Three channels sampling in parallel are used for the detection of $NO_2$ and organic nitrates, the three cavities are maintained at slightly above ambient temperature (305 K) to reduce density fluctuations. All three channels detect $NO_2$ (ambient, or produced by organic nitrate decomposition) using the absorption of $NO_2$ at 405 nm ($\sigma_{405nm}$ at 298 K $\approx 6 \times 10^{-19}$ cm$^2$ molecule (Voigt et al., 2002)). The cavity mirrors (Advanced Thin Films) have a reflectivity R $\approx 0.99995$ which results in an average



ring down of 30 - 35 µs at 700 mbar of dry synthetic air. The 405 nm cavities have a data acquisition system and optical setup which is similar to the 662 nm cavities, although an optical isolator to prevent back reflections was found unnecessary at 405 nm. To keep the peak emission wavelength at 405 nm, the laser diode is held at 35 °C. The maximum intensity coupled into each of the three 405 nm cavities is in the range 10 to 15 mW. The laser wavelength is monitored using a mini

spectrograph (Ocean Optics, type HR4C5451) which records diffuse back reflexions from the $\Sigma$ANs cavity input mirror. The effective cross-section for the three 405 nm channel during the calibrations and field deployment was 5.9 x $10^{-19}$ $cm^2$ $molec^{-1}$ (based on the laser emission spectrum and the $NO_2$ absorption cross-section).

Ambient $NO_2$ is detected directly when sampling from a room temperature inlet, whereas PNs and ANs are thermally decomposed in two heated sections of glass tubing at $\approx$ 448 K and $\approx$ 648 K and then detected as $NO_2$. In the 448 K channel

($\Sigma$PNs channel), the sum of ambient $NO_2$ plus $NO_2$ from the thermal decomposition of PNs is measured. In the 648 K channel ($\Sigma$ANs channel) $NO_2$ from the decomposition of ANs is additionally detected. The temperature set in the $\Sigma$PNs channel (448 K) was chosen by measuring the temperature dependence of the thermal dissociation of PAN ($\approx$ 2 ppbv) in synthetic air as provided by a diffusion source of PAN in tridecane. The normalized [$NO_2$] signal as a function of temperature (Fig. 4) shows that complete decomposition to $NO_2$ is achieved at a nominal temperature of $\approx$ 433 K. The

temperatures given in the text and in Fig. 4 correspond to the external surface of the heated inlets and not necessarily that of the gas flowing through them. PANs have a thermal stability which is largely independent of the R group (Roberts, 1990b; Kirchner et al., 1999). For the detection of atmospheric PNs we set the temperature to 15 K above the threshold measured in the laboratory for PAN, i.e. 448 K. Figure 4 also shows the temperature dependent decomposition of a sample of isopropyl nitrate in synthetic air: a yield of 100 % is reached at a set temperature of $\approx$ 633 K. As for PANs, the thermal decomposition

of all ANs is expected to be similar, so a working temperature of 648 K was chosen.

Rayleigh scattering of 405 nm light by air (Fuchs et al., 2009; Thieser et al., 2016) results in sensitivity of the ring-down constant to pressure changes and requires that the three 405 nm cavities are actively pressure regulated. For this reason, and because zeroing requires overfilling the inlet line (see below), the 405 nm channels have an independent inlet system. In total, $\approx$ 8 slm air is sampled through ¼" PFA piping of which 0.5 slm are directed to a digital humidity sensor. Active

pressure control of the cavities is achieved by pumping a regulated fraction of the main flow (usually $\approx$ 0.5 slm) through a flow controller directly to the exhaust system (see Fig. 2) to maintain the desired pressure (generally about 700 mbar). About 7 slm of ambient air is drawn through a filter held by a PFA filter holder. The particle filters used to protect the 405 nm cavities are the same as for the 662 nm cavities but are changed at a much lower frequency (once per day per hand) due the lower reactivity of $NO_2$ and organic nitrates to surfaces. After passing through the particle filter, air is then split into three

equal flows ($\approx$ 2.3 slm), resulting in residence times of 1.7 s, 1.3 s and 1.1 s for the $NO_2$, $\Sigma$PNs and $\Sigma$ANs channels, respectively. The three 405 nm channels consist of vertically mounted glass tubing (length $\approx$ 55 cm, ID = 12 mm) connected to the cavities by PFA fittings as described in Sect. 2. For the $\Sigma$PNs and $\Sigma$ANs channels, a portion of the glass tubing is wrapped with heating wire and heavily insulated to achieve the desired thermal dissociation temperatures of 448 and 648 K. The first 10 cm of the heated section of the glass tubing in the $\Sigma$PNs and $\Sigma$ANs channels is filled with glass beads ($\approx$ 0.5 mm





diameter, Sigma-Aldrich G9268) supported on a glass frit (circa 2 cm long). More detail and a diagram is given in the supplementary information. The surfaces of the glass beads and glass frit scavenge organic radicals formed in the thermal decomposition of PNs and ANs and thus reduce the impact of radical recombination with $NO_2$ and oxidation of NO, which can bias the results obtained, especially under conditions of high NO$x$ (Day et al., 2002; Paul and Osthoff, 2010; Thieser et

al., 2016). Details of the laboratory experiments carried out and the corrections necessary to take such processes into account are described in Sect. 3.2.

The three 405 nm channels are zeroed for 30 s every 4 minutes by overflowing the inlet with ≈ 0.2 slm of bottled, synthetic air or with scrubbed air provided by a zero-air generator (CAP 180, Fuhr GmbH). An upper limit to the $NO_2$ content of the zero air of ≈ 20 pptv was obtained by use of a blue-light converter as described previously (Thieser et al., 2016). Previous

work has shown that the different scattering coefficients for humid, ambient air and the dry synthetic air used to zero the instrument can lead to a bias in the measured [$NO_2$]. The difference in the Rayleigh scattering cross-section at wavelengths close to 405 nm between synthetic air and water ($\Delta\sigma$) is reported to be between 0.4 and 0.5 x $10^{-21}$ cm$^2$ molecule$^{-1}$ (Fuchs et al., 2009; Thieser et al., 2016). At 50 - 60 % humidity at 30 °C and 1 bar, this results in an offset of ≈ 200 pptv in the [$NO_2$] signal. Thieser et al. (2016) have indicated that the 20 % difference in the reported scattering cross-sections imply that this is

a significant source of uncertainty at high relative humidity and low $NO_2$. To avoid introducing additional error by correcting for this, we implemented a zero-air humidification system which enables the 7 slm air flow used for zeroing the instrument to be actively matched (to better than 2 % up to 80 % RH) to ambient relative humidity (RH of ambient and zero-air both measured at 0.2 Hz). This was achieved by passing a variable fraction of the zero air to a bubbler containing distilled water. Taking $\Delta\sigma$ = 0.5 x $10^{-21}$ cm$^2$ molecule$^{-1}$, the error arising for a 2 % humidity difference at room temperature and 1 bar

(measuring conditions of both humidity sensors) is equal to 10 pptv.

As the ring-down times are shorter at 405 nm (typically 35 μs) the diode laser modulation frequency could be increased so that 1666 ring-down constants were obtained per second. Figure 3 shows an Allan deviation plot (1σ) for the $NO_2$, ΣPNs and ΣANs channels. The 1σ standard deviation for a 1 s integration time for the 3 405 nm channels are 25, 40 and 45 pptv, respectively. The higher noise level associated with the ΣPNs and ΣANs channels results from slightly lower mirror

reflectivity (different batch) compared to the $NO_2$ cavity.

## 3 Data Corrections, precision and total uncertainty

When based on known absorption cross-sections, the CRDS method is essentially calibration free. However, a number of corrections are necessary to convert measured ring-down times into volume mixing ratios of absorbing trace gases. Some corrections are related to the physical construction of the cavities and others are related to chemical processes in the inlets

and cavities that can bias the concentrations measured. The size and accuracy of these corrections contribute significantly to the overall uncertainty of the measurements and are discussed in detail below. The more straightforward corrections needed are related to the effective absorption path length (*l*-to-*d* ratio), losses of $NO_3$ radicals in inlets and filters as previously





described (Schuster et al., 2009; Thieser et al., 2016) and formation of $NO_2$ during the sampling time through $O_3$ + NO reaction. For this instrument, the effective *l*-to-*d* ratio is $0.69 \pm 0.02$ for the channels detecting at 405 nm and $0.77 \pm 0.04$ for the channels detecting at 662 nm. Based on several laboratory experiments, the transmission of $NO_3$ in the filter / filter holder is $70 \pm 3$ %. No losses of $N_2O_5$ on passage through the filter holder are observed. The transmission of $NO_3$ in the $NO_3$ and $N_2O_5$ channels are $97.4 \pm 2.5$ % and $90.2 \pm 5$ %, respectively. Details and results of the experiments to derive *l*-to-*d* ratios and correction factors for $NO_3$ losses are given in the supplementary information. Corrections related to NO oxidation by $O_3$ are described in Sect. 3.2.5.

### 3.1 $NO_3$ and $N_2O_5$ channels

For averaging periods of longer than $\approx 1$ s, the detection limit of both $[NO_3]$ and $[N_2O_5]$ is defined mainly by variation / drift in the zero-signal. Drifts can arise from changes in the mirror reflectivity and through thermal and mechanical stresses that influence the cavity alignment. The detection limit can be estimated by the $2\sigma$ standard deviation of the difference from one zeroing period to the next one for the whole campaign. The values for the $NO_3$ and $N_2O_5$ channels are respectively 1.5 and 3 pptv and imply that 95 % of the time, the difference between one zero value to the next one is lower than 1.5 pptv and 3 pptv respectively. The $NO_3$ limit of detection is thus 1.5 pptv for 1 s averaging and zeroing every three minutes. As $N_2O_5$ is obtained by difference, the detection limit is $\approx 3.5$ pptv (also for 1 s averaging). The total uncertainty for the $[NO_3]$ measurement can be estimated by propagation of the individual uncertainties associated with the correction factors (see above) and the uncertainty associated with the effective absorption cross-section, which depends on the $NO_3$ absorption cross-section uncertainty ($\approx 10$ %, taken from (Osthoff et al., 2007)) and the uncertainty associated with measurement of the laser emission spectrum ($\approx 5$ %). The resulting, total uncertainty for $NO_3$ measurement is 25 %. The uncertainty associated with the $[N_2O_5]$ measurement depends on the absolute values of $[NO_3]$ and $[N_2O_5]$ and is therefore variable. As an example, combining the 25 % uncertainty on the total $NO_3$ measured in the $N_2O_5$ channel with $[NO_3]$ and $[N_2O_5]$ mixing ratios of 50 pptv and 500 pptv, respectively results in an uncertainty of 28 % for the $N_2O_5$ measurement.

### 3.2 $NO_2$, $\Sigma$PNs and $\Sigma$ANs channels

### 3.2.1 Losses of $NO_2$ in the heated $\Sigma$PNs and $\Sigma$ANs inlets

The transmission of $NO_2$ through the hot glass inlets (with frit and packed with glass beads) of the $\Sigma$PNs and $\Sigma$ANs channels was investigated by sampling $NO_2$ in synthetic air (up to 10 ppbv) simultaneously into the three 405 nm channels. $NO_2$ is depleted slightly in the hot inlets with a transmission of $98.5 \pm 2$ % and $95.5 \pm 2$ % found for the $\Sigma$PNs and $\Sigma$ANs inlets, respectively (see Fig. 5). The $NO_2$ transmission was measured before, during and after the field deployment of this instrument (corresponding to a period of several months) and found to be constant and also independent of relative humidity.



### 3.2.2 Reaction of radicals with NO / NO$_2$ in the ΣPNs channel (448 K)

The thermal decomposition of PNs leads to the formation of organic radicals and NO$_2$. In an ideal situation, in which PNs decompose 100 % to NO$_2$ and when NO$_2$ does not undergo any further production or loss reaction, the total [NO$_2$] measured in the ΣPNs channel is equal to the sum of ambient NO$_2$ and PNs. Under the operating conditions of the CRDS instrument

described here, the total [NO$_2$] signal in the ΣPNs channel can however be biased by a number of reactions initiated by the organic radicals. The influence of these reactions has been described in the literature (Day et al., 2002; Paul and Osthoff, 2010; Thieser et al., 2016). Taking the example of peroxyacetyl nitrate (PAN), subsequent to its thermal decomposition (R1a), there are processes that lead both to the removal of NO$_2$ (R1b and R5) and to its formation (R2 to R4).

$$CH_3C(O)O_2NO_2 + M \quad \rightarrow \quad CH_3C(O)O_2 + NO_2 + M \tag{R1a}$$
$$CH_3C(O)O_2 + NO_2 + M \quad \rightarrow \quad CH_3C(O)O_2NO_2 + M \tag{R1b}$$
$$CH_3C(O)O_2 + NO\ (+\ O_2) \quad \rightarrow \quad NO_2 + CH_3O_2 + CO_2 \tag{R2}$$
$$CH_3O_2 + NO\ (+\ O_2) \quad \rightarrow \quad HCHO + HO_2 + NO_2 \tag{R3}$$
$$HO_2 + NO \quad \rightarrow \quad OH + NO_2 \tag{R4}$$
$$OH + NO_2 \quad \rightarrow \quad HNO_3 \tag{R5}$$

Reactions R2 to R4 show that, in the most unfavourable scenario, if all organic radicals react with NO, each peroxyacetyl radical can lead to the formation of 3 NO$_2$ molecules, biasing the result by the same factor. Likewise, the presence of very high NO$_2$ could conceivably result in complete reformation of PAN (R1b). The size and sign of the bias thus depends on the

relative concentrations of NO and NO$_2$ and, most importantly, on the rate of loss of organic radicals to the reactor walls or via thermal decomposition. The bias resulting from these reactions can be reduced by minimising the residence time between thermal dissociation and detection and making pressure dependent recombination reactions inefficient by working at low pressures e.g. by using LIF detection of NO$_2$ (Day et al., 2002; Wooldridge et al., 2010). As CRDS instruments operate at higher pressures to maintain sufficient sensitivity (typically from 0.5 to 1 bar) we have taken a different approach and

optimised the surface losses of the organic radicals by modifying the surface-to-volume-ratio in the heated inlets.
First experiments on PAN samples using glass wool as a radical scavenger resulted in the desired reduction in the rate of recombination of CH$_3$C(O)O$_2$ with NO$_2$. Glass wool was however observed to greatly enhance (rather than reduce) the oxidation of NO to NO$_2$. This is presumably the result of a surface catalysed process as previously observed on powdered, aluminium-silicate mineral dust samples (Hanisch and Crowley, 2003) and may be related to the formation of oxidised

surface sites that can react with NO. This lead us to test glass beads, which presumably have less reactive "defect" sites than glass wool but sufficient surface area to remove a large fraction of the organic radicals (see below). We conducted a series of experiments with different mixtures of PAN and NO$_2$/NO (from 1 ppmv in synthetic air gas bottles) and analysis of the observations by numerical simulation similar to that presented in Thieser et al. (2016).



The set of chemical reactions used in the numerical simulations is essentially the same as that described in Thieser et al. (2016). One exception is formation of $NO_2$ via the reaction between $O_3$ and NO which Thieser et al. (2016) treated as occurring independently of other chemical processes. Here, this reaction treated more rigorously by including it in the chemical simulation. The other difference is the use of two different values for a rate constant ($k_s$) used to calculate the

kinetic limitation on the radical uptake coefficient ($\gamma$, see below). In order to simulate all laboratory data sets different values of $k_s$ (one for section A and B and one for section C and D, see Fig. S1) are required to account for the temperature gradient in the heated sections of the inlets. The heterogeneous loss of radicals to the reactor walls is based on a Langmuir-Hinshelwood mechanism, with the first-order rate constant ($k_w$) given by Eq. (2).

$$k_w = \frac{\gamma \bar{c} A}{4} \qquad\qquad\qquad \text{Eq. (2)}$$

The uptake coefficient, $\gamma$, depends on both kinetic and diffusive limitations as described in detail by (Thieser et al., 2016). Here we focus on the role of glass beads in enhancing the rate of uptake and reaction of organic radicals by increasing the surface area available for reaction ($A$). Without the glass beads, the value of $A$ in the heated inlets is 3.5 cm$^2$ cm$^{-3}$

(corresponding to a cylinder with internal diameter 1.2 cm. The presence of the glass beads, with an average diameter of 0.5 mm, results in a surface-to-volume ratio of $\approx 100$ cm$^2$ cm$^{-3}$. The exact determination of the temperature profile in the inlet is possible in the case of a classic cylindrical reactor but is difficult to achieve with the presence of the glass beads and glass frit. An approximate profile in section D of the reactor (see Fig. S1) was obtained by measuring the temperature between the fritted glass and the entrance of the cavity. It is assumed that the temperature in section B rises from ambient to 448 K at the

point where the gas enters section C (corresponding to the fritted glass section). This profile is also used to calculate the residence time in the ΣPNs channel of 1.3 s. To take into account the variations of temperature, pressure, specific surface area and diffusion radius along the glass reactor, a profile for each of those parameters is used as input in the numerical simulation.

Figure 6a shows results of a set of 4 experiments in which different amounts of PAN are mixed with different amounts of

$NO_2$ and monitored in the 3 channels. The Y-axis shows the total $[NO_2]$ measured in the ΣPNs channel ($[NO_2]_{448K}$) minus $NO_2$ measured in $NO_2$ channel ($[NO_2]_{Amb}$). The negative slopes of these data indicate that $NO_2$ from PAN decomposition is lost at high mixing ratios of added $NO_2$. A similar set of data, but with added NO rather than $NO_2$ is displayed in Fig. 6b. In this case, the positive slopes indicate that $NO_2$ is being formed from reaction of organic radicals with NO. Note that the solid lines are the results of the numerical simulations (with identical mechanism) of $NO_2$ formation and loss in both experiments.

The mixing ratios of PAN listed are those required in the model to match the experimental data. In both datasets the slopes are weakest at low PAN concentrations, reflecting the fact that radical-NO or radical-$NO_2$ reaction rates will be dependent on the radical concentrations. The mixing ratios of PAN cannot be derived by simple back extrapolation to zero $NO_2$ or NO as



this does not take into account the $NO_2$ formed when PAN itself decomposes. At the highest mixing ratios of $NO_2$, up to 25% of the peroxyacetyl radical recombines with $NO_2$ while the other 75% is lost to the walls or did not react with $NO_2$. The analysis of the measurements shows that (without correction) $NO_2$ would be overestimated by approximately 40 % for 3 ppb of PAN when sampling air containing 8 ppbv of NO. This is much lower that the ≈ 150 % reported by Thieser et al.

(2016) for similar conditions in their ΣPNs channel, reflecting the increased loss of radicals on the glass surfaces.

In both series of experiments, the model reproduces the data for the whole range of [PAN], [$NO_2$] and [NO] explored. A set of experiments was carried out to investigate whether the formation (or loss) of $NO_2$ described above could be influenced by the presence of water vapour acting e.g. as a quencher of a surface reaction. A mixture of PAN plus $NO_2$ was humidified to 60 % RH with other operating conditions similar to the previous experiments. The numerical model reproduced the

measurement data without the need to modify the wall loss rates or add any chemical processes involving $H_2O$, suggesting that the gas- and surface reactions taking place are not significantly influenced by adsorbed water at these temperatures. In order to correct field data for these biases, an iterative fitting procedure is used and described in detail in Sect. 4.2.

### 3.2.3 Reaction of radicals with NO / $NO_2$ in the ΣANs channel (648 K)

The higher temperature (648 K) in the ΣANs channel changes the chemical processes substantially compared to 448 K. ANs

can now decompose to $NO_2$ and an alkyl radical fragment whereas the peroxyacetyl radical (from PAN decomposition) is thermally unstable. The $NO_2$ generated in the ΣANs channel when adding $NO_2$/NO to PAN samples is displayed in Fig. 6c and 6d. The results show a greatly reduced effect of $NO_2$ recombination or NO oxidation compared to the ΣPNs channel (Fig. 6a and 6b), largely resulting from the instability of the of $CH_3C(O)O_2$ radical. The model takes into account two main pathways for the fate of the peroxyacetyl radical (R6 and R7) in which the acetyl radical ($CH_3CO$) formed in reaction R6 can

react further with $O_2$ to reform a peroxyacetyl radical or an OH radical and a α-lactone (R8 and R9). $CH_2C(O)OOH$ can also decompose to OH and an α-lactone (R10). If not lost to the walls, OH can react with $NO_2$ to form $HNO_3$ according to reaction (R5).

$$CH_3C(O)O_2 + M \quad\rightarrow\quad CH_3CO + O_2 + M \tag{R6}$$

$$CH_3C(O)O_2 \quad\rightarrow\quad CH_2C(O)OOH \tag{R7}$$

$$CH_3CO + O_2 + M \quad\rightarrow\quad CH_3C(O)O_2 + M \tag{R8}$$

$$CH_3CO + O_2 \quad\rightarrow\quad OH + C_2H_2O_2 \tag{R9}$$

$$H_2C(O)OOH \quad\rightarrow\quad OH + C_2H_2O_2 \tag{R10}$$

$$CH_3CO + (O_2) \quad\rightarrow\quad CH_3O_2 + CO \tag{R11}$$

The acetyl radical can also form a methylperoxy radical $CH_3O_2$ according to reaction R11. In the presence of NO the peroxy radicals ($CH_3C(O)O_2$, $CH_3O_2$ and $HO_2$), if not lost to the glass surface, can lead to the formation of $NO_2$. To simulate the reactions in the ΣANs channel, the temperature profile was determined in the same way as the ΣPNs channel profile. Due to



the higher gas flow velocity at the elevated temperatures, the pressure profile is slightly different than in the ΣPNs channel and gives a cavity pressure that is about 10 mbar lower. The results of the experiments involving addition of $NO_2$ to PAN samples in the ANs channel are presented in Fig. 6c. For the range of PAN concentrations covered, the loss of $NO_2$ by chemical recombination is about 5% of the initial PAN. The initial PAN concentration used as input variable for the numerical simulations is, for all 4 experiments, about 3 to 5 % higher than that needed to fit the data obtained in the PNs channel (Fig. 6a), which is most probably related to model uncertainties. Since this 3 - 5% difference is the same for all four experiments and is apparently not correlated with the amount of PAN, it is added to the calculation of total uncertainty for the PNs measurement. The results of the NO addition experiments are shown in Fig. 6d. The thermal decomposition of the peroxyacetyl radical reduces the positive bias due to NO oxidation so that a maximum factor of only 1.04 (compared to 1.40 in the ΣPNs channel) is obtained. The initial PAN concentration required to fit the NO addition dataset at 648 K are 5 to 10 % higher than at 448 K which is related to uncertainty in the [NO] mixing ratio used or a bias in the simulation.

In order to investigate the role of organic (alkyl) radicals generated from the thermal decomposition of ANs, a set of experiments was conducted in which a sample of $i$-propyl nitrate ($C_3H_7ONO_2$) in synthetic air was mixed with different amounts of NO and $NO_2$. Thermal decomposition of $i$-propyl nitrate (R12) is followed by reactions R13 and R14 which generate $HO_2$ and $CH_3O_2$, which can convert NO to $NO_2$, and sequester $NO_2$ as $HO_2NO_2$ and $CH_3O_2NO_2$

$$C_3H_7ONO_2 \rightarrow C_3H_7O + NO_2 \tag{R12}$$
$$C_3H_7O + O_2 \rightarrow CH_3C(O)CH_3 + HO_2 \tag{R13}$$
$$C_3H_7O + M \rightarrow CH_3 + CH_3CHO + M \tag{R14}$$

Figure 7a and 7b shows that the addition of NO increases the amount of $NO_2$ formed per AN, whereas the presence of $NO_2$ results in a negative bias to the data. These effects are captured well for both datasets by the model simulations (blue lines).

### 3.2.4 Effect of thermal decomposition of $O_3$

At high temperatures, $O_3$ decomposes according to reaction R15. Although most of the O atom produced reacts with $O_2$ to reform ozone (R16), it can form NO + $O_2$ in the presence of $NO_2$ (R17).

$$O_3 \rightarrow O_2 + O \tag{R15}$$
$$O + O_2 + M \rightarrow O_3 + M \tag{R16}$$
$$O + NO_2 \rightarrow O_2 + NO \tag{R17}$$

The importance of this process is strongly dependent on the operating conditions of the instrument, recent studies showing that higher temperatures or pressures may result in a significant, negative bias from the ozone pyrolysis initiated reduction of

NO$_2$ (Lee et al., 2014; Thieser et al., 2016). Assuming steady state for the O atom concentration in the heated inlets, the loss of NO$_2$ can be approximated as follows:

$$ -d[NO_2] = [NO_2][O_3]\left(\frac{k_{15}k_{17}}{k_{16}[O_2]}t\right) $$

Eq. (3)

The term in brackets is constant at constant temperature, pressure and flow rate through the heated inlet and can be determined experimentally by sampling different mixtures of NO$_2$ and O$_3$ simultaneously through the ambient temperature channel and the two "hot" channels. Figure 8 shows the difference $[NO_2]_{Amb} - [NO_2]_{648K}$ (in ppbv) as a function of $[NO_2] \times [O_3]$ for three different levels of ozone (50, 115 and 185 ppbv). A linear fit through the data points yield a slope of 2.54 ±

0.26 x 10$^{-4}$ ppbv$^{-1}$, which is < 50 % of the value found by (Thieser et al., 2016). The difference can be attributed mainly to the use of a lower oven temperature (648 K in this work instead of 723 K). No reduction of NO$_2$ is observed in the ΣPNs channel because thermal decomposition of O$_3$ at 448 K is too slow. To illustrate the impact of the thermal dissociation of O$_3$ in the ANs channel we note that 5 ppbv of total NO$_2$ in the presence of 50 ppbv of O$_3$ results in the removal of 75 pptv NO$_2$. Since the amount of O$_3$ decomposed and NO produced in this process is negligible compared to ambient amounts it does not

affect the input conditions for the numerical simulations and can therefore be treated separately using Eq. (3).

### 3.2.5 Effect of NO oxidation by O$_3$

A positive bias to the measurement of NO$_2$ may result from NO oxidation by O$_3$ during the time it takes for the sampled air to flow through the inlets and cavities of the instrument (R18).

20                      NO + O$_3$                     →      NO$_2$                                                         (R18)

The rate coefficient of this reaction is temperature dependent and the effect is larger at 448 K and 648 K than at room temperature. To investigate this interference, different mixtures of O$_3$ and NO were sampled simultaneously through the 3 channels. The results are shown in Fig. 9 where data from four experiments with different O$_3$ mixing ratios (41, 62, 80 and

131 ppbv) are plotted. The solid lines are fits to the data, the slopes of which are second order rate constants multiplied by a reaction time (cm$^3$ molecule$^{-1}$). The rate constant for the gas-phase reaction between NO and O$_3$ is listed as $k_{O3+NO}$ = 9 x 10$^{-19}$ x exp(-850/T) × T$^{2.25}$ cm$^3$ molecule$^{-1}$ s$^{-1}$ (Atkinson et al., 2004). For the ΣPNs channel (448 K), this expression results in an underestimation of the effective (measured) production of NO$_2$ by a factor 1.06 and 1.52 for the ΣANs (648 K) channel. The most likely reason for this discrepancy is that some NO is oxidised in a surface catalysed process, presumably involving

surface sites that are activated following interaction with O$_3$. To account for this extra source of NO$_2$, the O$_3$ plus NO reaction is implemented in the chemical model (with modified rate expressions) for the 448 K and 648 K data corrections (see Sect. 3.2.2 and 3.2.3) and corrected manually for the NO$_2$ channel.



### 3.2.6 Detection of other species as $NO_2$ after thermal dissociation

Atmospheric trace gases other than PNs and ANs can potentially be detected as $NO_2$ in this instrument. As thermal decomposition of $N_2O_5$ to $NO_3$ and $NO_2$ has been shown to be 100 % efficient in our 383 K inlet in the 662 nm channels, we expect 100 % dissociation at the higher temperatures of the $\Sigma$PNs and $\Sigma$ANs channels. This may represent a significant

source of bias during the night when $N_2O_5$ mixing ratios can be large. However, as the 5-channel instrument simultaneously monitors $N_2O_5$, this can be easily corrected. Here we consider possible $HNO_3$ and $ClNO_2$ interferences in the $\Sigma$ANs channel. $HNO_3$ is a major reservoir of tropospheric NO$x$. Although $HNO_3$ is thermally stable at temperatures below 700 K, Wild et al. (Wild et al., 2014) report $\approx$ 95 % conversion of $HNO_3$ to $NO_2$ at 648 K whereas (Thieser et al., 2016) found about 10 % at 723 K. In order to test for unwanted detection of $HNO_3$ when sampling from the hot inlets (with glass beads) in the 5-

channel instrument, we injected a sample of $HNO_3$ in synthetic air directly prior to the flow division before entering the three 405 nm channels. In this way possible losses of $HNO_3$ in the inlet in front of the heated sections of the instrument were minimised. $HNO_3$ was generated in a custom built permeation source, which was calibrated by absorption spectroscopy at 184.85 nm, where $HNO_3$ absorbs strongly. The permeation source generated several ppmv of $HNO_3$ in a flow of 10 sccm, which, following dilution, provided a mixing ratio of $\approx$ 30 ppbv at the CRDS inlet. The permeation source also generated

about 3 ppbv of $NO_2$. Figure 4 (open circles) shows that $NO_2$ arising from $HNO_3$ decomposition could not be observed in either heated inlet enabling us to set an upper limit to the decomposition efficiency of $HNO_3$ to $NO_2$ of < 0. 5 %. This is in broad agreement with the previous measurement by (Thieser et al., 2016) who saw no evidence for $HNO_3$ decomposition below 650 K and is in stark contrast to that reported by Wild et al. (Wild et al., 2014). While we have no rigorous explanation of this, we note that the gas-phase thermal dissociation of $HNO_3$ to $NO_2$ at the temperature of these experiments

is too slow to explain its formation, which suggests that it is probably surface catalysed in the experiments of Wild et al. (2014), implying that different glass-types or chemical history of the surface may influence the formation of $NO_2$ significantly.

As reported previously, $ClNO_2$, formed in the atmosphere in the reaction between $N_2O_5$ and chloride containing particles, can be detected as $NO_2$ in TD-CRDS instruments (Thaler et al., 2011; Wild et al., 2014; Thieser et al., 2016) The $ClNO_2$

decomposition efficiency as a function of the oven temperature is plotted in Fig. 4. The $ClNO_2$ sample was generated by flowing $Cl_2$ in synthetic air over sodium nitrite crystals. In normal operating conditions (at 648 K), 90 $\pm$ 3 % of the $ClNO_2$ is decomposed to $NO_2$. As $ClNO_2$ mixing ratios are highly variable and can approach ppbv levels, $ClNO_2$ may thus represent a serious limitation to ANs measurements unless independent measurements are available to enable correction.

### 3.2.7 Detection limit and total uncertainty for PNs and ANs

The detection limit for the 405 nm channels can be estimated in a similar manner to that described for the 662 nm channels. The 2$\sigma$ standard deviation for consecutive zeros for the $NO_2$, $\Sigma$PNs and $\Sigma$ANs channels are respectively 59 pptv, 74 pptv and 54 pptv. Using those values, we obtain detection limits for [$NO_2$], [PNs] and [ANs] of 59 pptv, 94 pptv and 80 pptv. The





detection limits for [PNs] and [ANs] are obtained by error propagation on the $NO_2$ and PNs channels and $NO_2$ and ANs channel (see Sect. 4).

The total uncertainty associated with the [$NO_2$] measurement is a combination of the uncertainties in the *l*-to-*d* ratio, humidity matching of the zero and ambient air, the correction for the $NO + O_3$ reaction and on the absorption cross-section.

The total uncertainty on the absorption cross-section is estimated to 6 % taking into account the error on the reference cross-section (Voigt et al., 2002) and fluctuation in the laser emission spectrum. This value, when combined with the uncertainty associated with the *l*-to-*d* ratio, results in 6.5 % uncertainty. As mentioned in Sect. 2.2, the upper limit for bias caused by humidity matching errors when zeroing is 10 pptv, which give a final value for [$NO_2$] of 6.5% + 10 pptv. The uncertainty associated with the correction for $NO_2$ formation in the $NO + O_3$ reaction depends on ambient ozone and NO levels.

Considering 10 % uncertainty on $k_{O3+NO}$ and a 5 % on [$O_3$] and [NO] and [$NO_2$], [NO] and [$O_3$] ambient levels of 1 ppbv, 0.5 ppbv and 50 ppbv, we obtain an uncertainty for this correction of 0.1 % on the final [$NO_2$] value.

The uncertainty on the total $NO_2$ detected in the ΣPNs and ΣANs channels is a combination of the same uncertainties involved in the [$NO_2$] plus the uncertainty on the $NO_2$ transmission through the glass beads. Taking the values mentioned in Sect 3.2.1, we get an uncertainty for total $NO_2$ detected of 7 % pptv for both ΣPNs and ΣANs channels. To correct for the

radical plus NO*x* biases, the total $NO_2$ signal measured in the two "hot" channels are used to constrain the chemical model described earlier. The uncertainty rising from those calculations depends on the total [$NO_2$] measured in each channel and is thus highly variable. Note that uncertainty arising from the humidity difference is cancelled out by the subtraction to obtain [PNs] and [ANs]. The uncertainty added by the calculation of the ambient [PNs] and [ANs] depends on the model uncertainties as well as on the uncertainty associated with the trace gas concentrations involved in this calculation. The

uncertainty associated with [ΣPNs] and [ΣANs] is related to ambient [$NO_2$], [$O_3$] and [NO]. The fact that the corrections rely on PAN and IPN chemistry led Thieser et al. (2016) to estimate the uncertainty associated with the model derived corrections to be 30 % (max) of the overall correction factor. The final uncertainties on the [ΣPNs] and [ΣANs] value depend on the total $NO_2$ signal in all three channels and can vary significantly. Below, we discuss the correction factors required for the $NO_2$, ΣPNs and ΣANs datasets obtained during the first field deployment of the instrument.

**4 First field deployment of the 5-channel CRDS**

The instrument described here was first deployed during the NOTOMO campaign (Nocturnal Observations at the Taunus Observatory; insights into Mechanisms of Oxidation) that took place during the summer 2015. The site, previously described in detail (Crowley et al., 2010; Sobanski et al., 2016), is situated on top of the Kleiner Feldberg Mountain (≈ 850 m ASL) at the southern limit of the forested Taunus mountain range and north of the Frankfurt/Mainz/Wiesbaden agglomeration. The

set of instruments deployed during NOTOMO was located in two research containers and sampled from a common inlet. Approximately ≈ 10 $m^3$ $min^{-1}$ air was drawn from a height of 8 m above ground through a 15 cm diameter stainless steel pipe connected to an industrial fan. Gases were sampled (≈ 23 slm) in the TD-CRDS (located in the upper container) from the



centre of the high flow inlet via a ½" PFA tubing (for the two 662 nm channels) and a ¼" PFA tubing (for the three 405 nm channels) ≈ 3.5 m from the top of the inlet. Residence times in the high flow inlet, in the ½" PFA tubing and the ¼" PFA tubing were 0.3, 0.05 and 0.1 s, respectively. In the following sections we shall not attempt to perform a detailed analysis of the entire campaign dataset, which is beyond the scope of this manuscript, but provide a more qualitative description of the

results and indicate the size of corrections applied and the total uncertainty.

## 4.1 Mixing ratios of $NO_2$, $NO_3$, $N_2O_5$

Figure 10a shows the measured mixing ratios of $NO_2$, $NO_3$, $N_2O_5$, $\Sigma PNs$ and $\Sigma ANs$ for one week between the 30[th] June and 7[th] July. During this period [$NO_2$] varied from 0.3 to 6 ppbv with an average value of 1.8 ppbv. $NO_3$ and $N_2O_5$ mixing ratios reach up to 40 pptv and 400 pptv at night and are below the detection limit during the day. These values are in the range of

previous measurements at this location (Crowley et al., 2010; Sobanski et al., 2016) in which $NO_2$, $NO_3$, and $N_2O_5$ mixing ratios of up to 20 ppbv, ≈ 200 pptv, ≈ 3000 pptv and, respectively were measured. Within a few minutes after sunset $NO_2$, $NO_3$ and $N_2O_5$ acquire thermal equilibrium and  the equilibrium constant, $K_{eq}$ can be calculated from [$N_2O_5$]/[$NO_2$][$NO_3$]. In Figure 11 we compare temperature dependent values of the equilibrium constant using measured concentrations of $NO_2$, $NO_3$ and $N_2O_5$ as well as values from the literature which are based on laboratory measurements (Atkinson et al., 2004;

Burkholder et al., 2016). Within combined uncertainty the values agree, giving us confidence in the accuracy of the measurements and the assessment of their uncertainty.

## 4.2 Mixing ratios of $\Sigma PNs$ and $\Sigma ANs$

By simply subtracting [$NO_2$] measured by the $NO_2$ channel to the total $NO_2$ signal measured by the PAN channel and the total $NO_2$ signal in the $\Sigma PNs$ channel from the total $NO_2$ signal in $\Sigma ANs$ channel we obtain uncorrected mixing ratios for

$\Sigma PNs$ and $\Sigma ANs$, which are plotted (black data points) in Fig. 10. In this context, uncorrected means that no chemical corrections have been applied (e.g. for re-formation of PNs in the presence of $NO_2$, or formation of $NO_2$ from NO oxidation with $O_3$ see Sect. 3) but physical corrections (e.g. $l$-to-$d$ ratio) have been made.

The procedure used to correct the [$\Sigma PNs$] and [$\Sigma ANs$] mixing ratios for the effects described in Sect. 3 utilises an iterative algorithm based on the chemical model used to simulate the laboratory data described in Sect. 3.2. To correct the $\Sigma PNs$ data,

a simulation using [$NO_2$], [$O_3$] and [NO] values and an initial guess for [$\Sigma PNs$] was performed. Values of $O_3$ were taken from ambient measurements whereas, in the absence of NO measurements, we assumed that its mixing ratio was zero at night-time and during day could be calculated from its photochemical steady state, [NO]$_{ss}$ = $J_{NO2}$[$NO_2$] / $k_{NO + O3}$ [$O_3$] where $J_{NO2}$ is the photolysis frequency of $NO_2$ (measured using a spectral radiometer) and $k_{NO + O3}$ is the rate constant for reaction of NO with $O_3$. This method of estimating [NO] resulted in satisfactory agreement with measurements from the environment

and geological agency of the state of Hessen (HLUG) for periods when NO was above the detection limit (> 1ppb) of their instrument, which monitors NO permanently at the site.



In the iterative procedure, [ΣPNs] was tuned automatically until the simulated total [NO$_2$] matched the measured [NO$_2$] in the ΣPNs channel. This procedure was applied to each data points (10 min resolution) obtained during the campaign. The corrected [ΣPNs] were used as an additional input to correct the [ΣANs] data. In this case, constant values for [NO$_2$], [O$_3$], [NO$_2$], [ΣPNs] and a variable value for [ΣANs] are inputted into the model to match the total NO$_2$ signal measured in the ΣANs channel. For the correction of the [ΣPNs] data, ambient N$_2$O$_5$ was taken into account by adding its mixing ratio to the initial [NO$_2$] values. This was also done also for the [ΣANs] correction. Measured ClNO$_2$ mixing ratio were decreased by the factor 0.9 to account for the decomposition efficiency and then added to the initial [NO$_2$] for simulation of the total NO$_2$ in the ΣANs channel. The corrected data are displayed as the red data points in Fig. 10a.

The overall correction factor (corrected data / uncorrected data) required for the ΣPNs and ΣANs measurements are illustrated in Fig. 10b and 10c. The peak of the distributions are at ≈ 1.1 indicating firstly that the corrections required are dominated by NO$_2$ rather than NO, and secondly that the corrections needed are rather small.

## 5 Conclusions

We have constructed and characterised a 5-channel, thermal dissociation cavity-ring-down spectrometer for measurement of the reactive nitrogen traces gases, NO$_2$, NO$_3$, N$_2$O$_5$ peroxy nitrates and alkyl nitrates that provide insight into the coupling of atmospheric, day- and night-time NOx and ROx chemistry. The total measurement uncertainties and limits of detection are estimated at 25 % and 2 pptv for NO$_3$, 28 % and 10 pptv for N$_2$O$_5$ and 6.5 % + 10 pptv and 59 pptv for NO$_2$.

Chemical interferences in the measurements of PNs and ANs were reduced by the use of low TD temperatures, and use of glass beads as radical scavenger. In combination with extensive laboratory tests to enable accurate correction for bias in PNs and ANs measurements resulting from reactions of organic radicals formed in the thermal dissociation channels this reduces overall uncertainty and extends the NOx regime in which the instrument can be operated. The total uncertainty associated with the PNs and ANs measurements is dependent on NOx levels and also on the presence (mainly at night) of ClNO$_2$ and N$_2$O$_5$. During the PARADE campaign the average correction was (1.1 ± 0.3) where the quoted uncertainty is an estimate of systematic error related to use of a numerical model to simulate chemical processes in the TD sections of the instrument.

**Acknowledgements**

We are grateful to the following: Bernard Brickwedde for technical/software assistance. We thank DuPont for provision of a sample of the FEP used to coat the cavity walls. We thank Heinz Bingemer for logistical support and use of the facilities at the Taunus Observatory during the NOTOMO campaign.



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



**Figure 1:** Atmospheric sources and sinks of the $NO_y$ species (in red circles) measured by the 5-channel TD-CRDS instrument described in this work and their link to VOC degradation and photochemical $O_3$ formation.





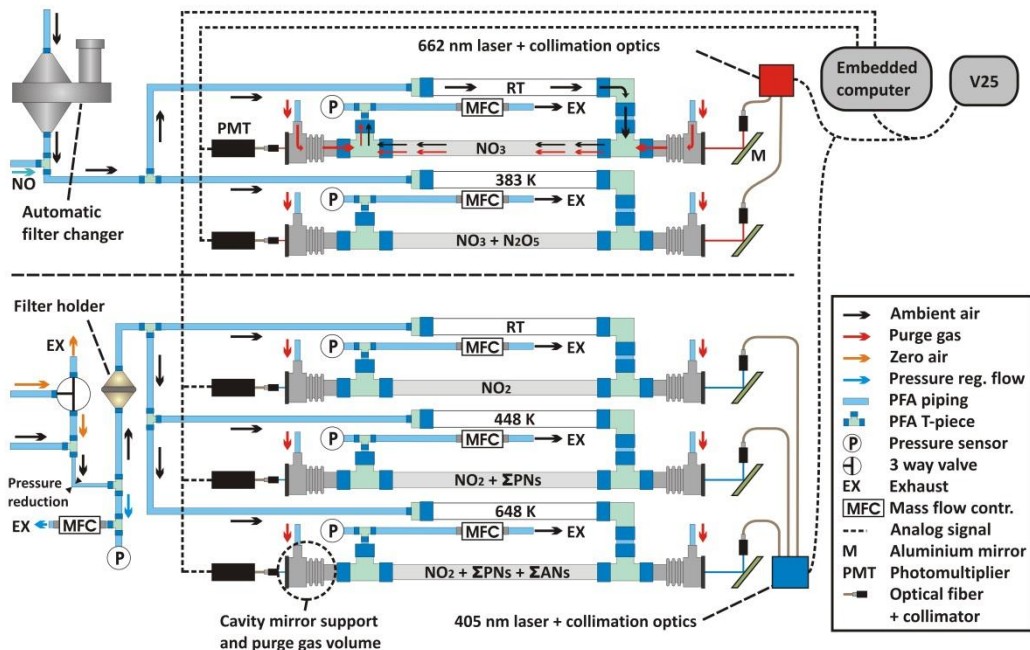

**Figure 2:** Schematic of the 5 channel TD-CRDS. Gas flows are represented by coloured arrows. The wall temperatures of the different inlets are given for each channel. RT = room temperature.



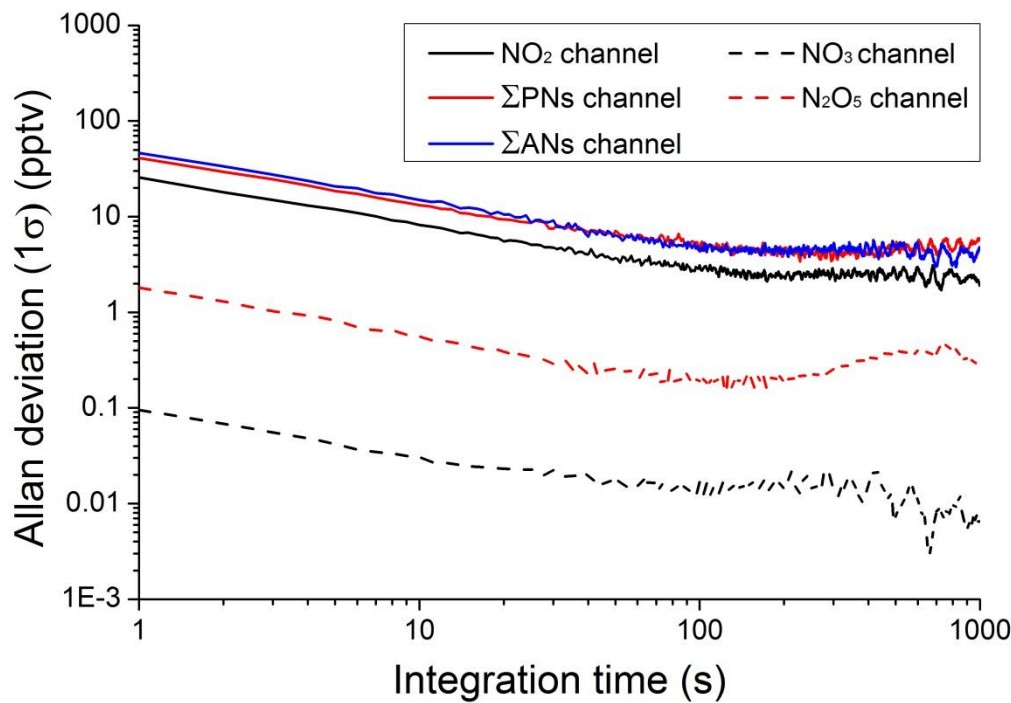

**Figure 3:** Allan deviation plot (1σ) for each of the five CRDS channels. The data were obtained in the laboratory at ≈ 298 K and a cavity pressure of 800 mbar.





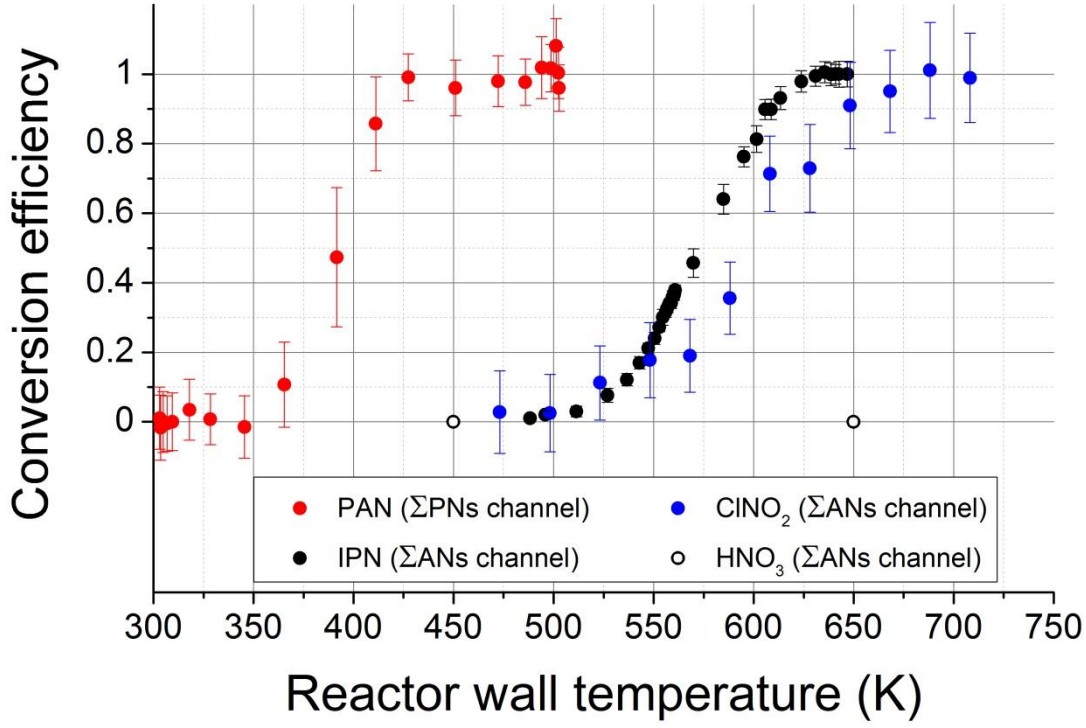

**Figure 4:** Decomposition profiles of PAN (red points), *i*-propyl nitrate (black points) and $ClNO_2$ (blue points) with statistical errors (1σ). The PAN decomposition profile was measured in the ΣPNs channel, the *i*-propyl nitrate (IPN) and $ClNO_2$ profile were measured in the ΣANs channel. The black circles indicate the lack of $HNO_3$ decomposition when sampled into the ΣANs channel.





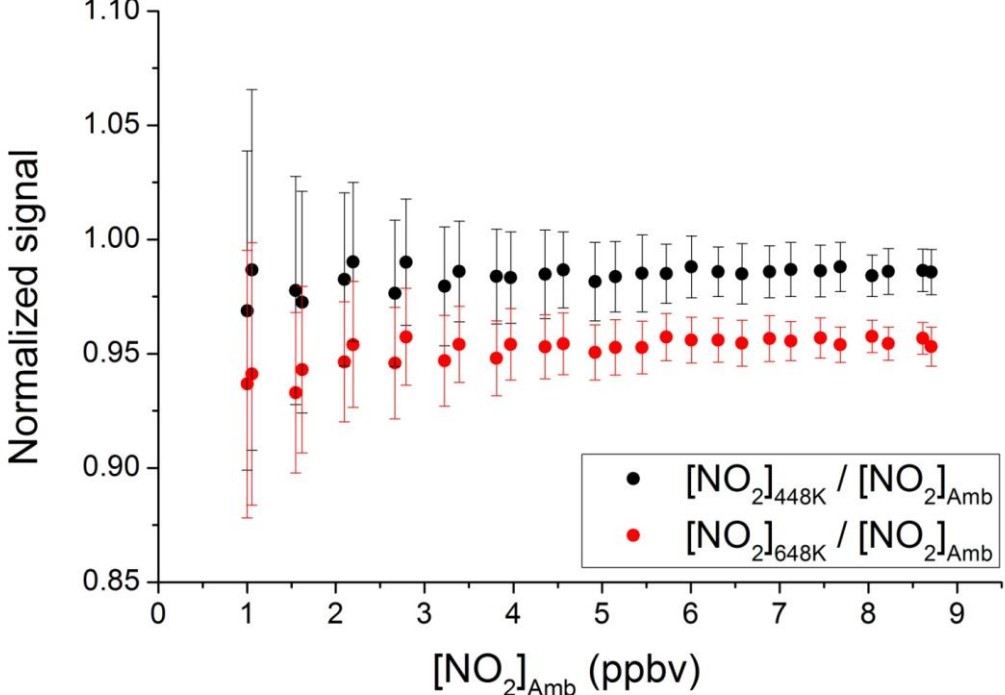

**Figure 5:** $NO_2$ loss in the heated inlets of the $\Sigma PNs$ and $\Sigma ANs$ channels. Black points represent the $NO_2$ mixing ratio measured in the $\Sigma PNs$ channel ($[NO_2]_{448K}$) normalized to the $NO_2$ mixing ratio measured in the $NO_2$ channel ($[NO_2]_{Amb}$). Red points represent the $NO_2$ mixing ratio measured in the $\Sigma ANs$ channel ($[NO_2]_{648K}$) normalized to the $NO_2$ signal measured in the $NO_2$ channel.





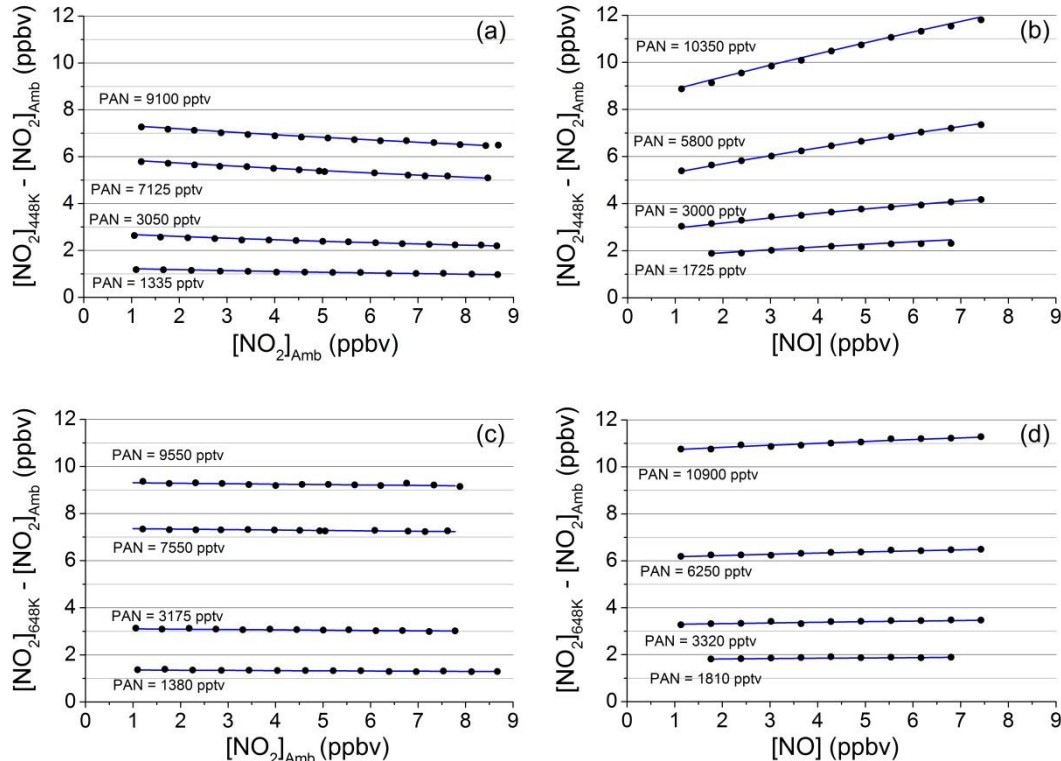

**Figure 6:** Modelled (lines) and measured difference between the $NO_2$ signal in the hot channels and ambient temperature channel for different PAN samples and different added amounts of NO and $NO_2$. **(a)** Addition of $NO_2$ to the 448 K (PNs) channel. **(b)** Addition of NO to the 448 K (PNs) channel. **(c)** Addition of $NO_2$ to the 648 K (ANs) channel. **(d)** Addition of $NO_2$ to the 648 K (ANs) channel.





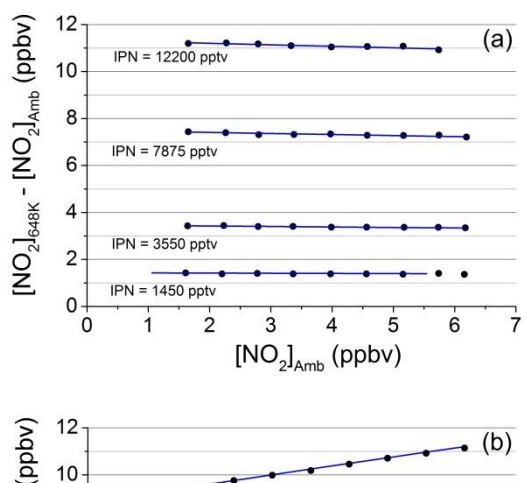

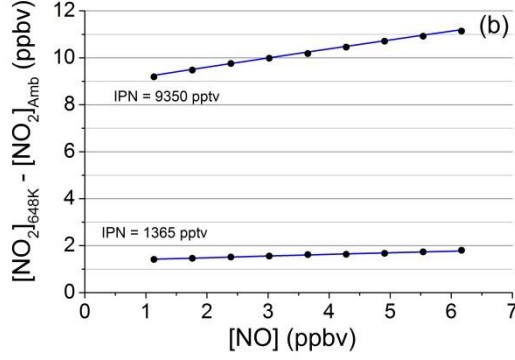

**Figure 7:** Modelled (lines) and measured difference between the $NO_2$ signal in the 648 K channels and ambient temperature channel for *i*-propyl nitrate (IPN) samples and different added amounts of $NO_2$ **(a)** or NO **(b)** .





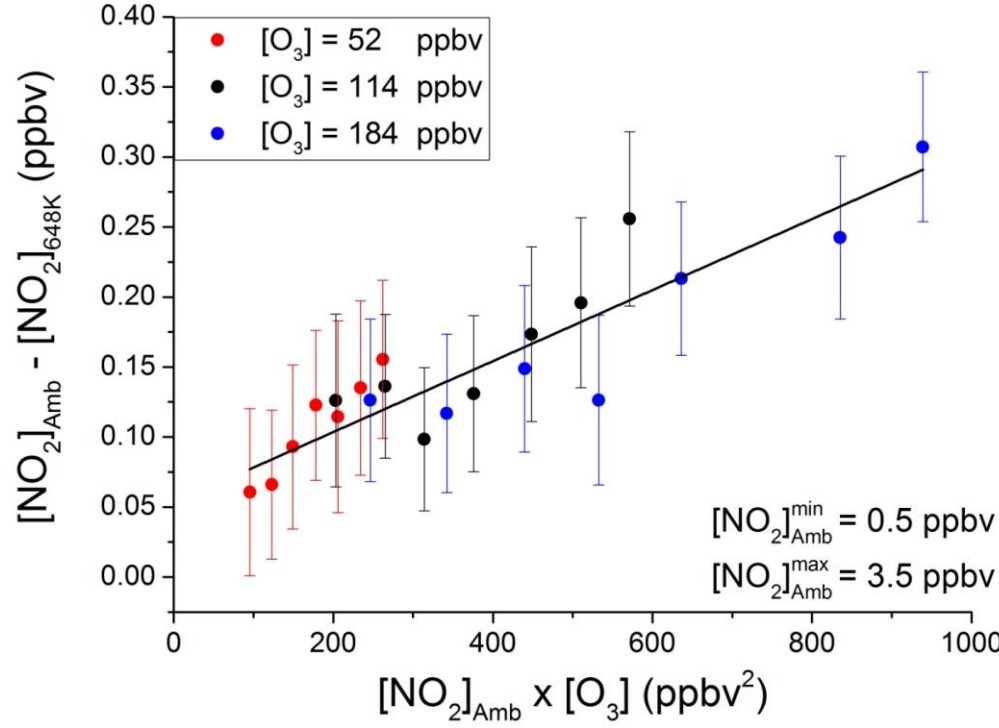

**Figure 8:** Influence of $O_3$ pyrolysis. Loss of $NO_2$ (in ppbv) in the heated $\Sigma$ANs channel compared to the $NO_2$ channel versus the product of the $NO_2$ and $O_3$ mixing ratios.





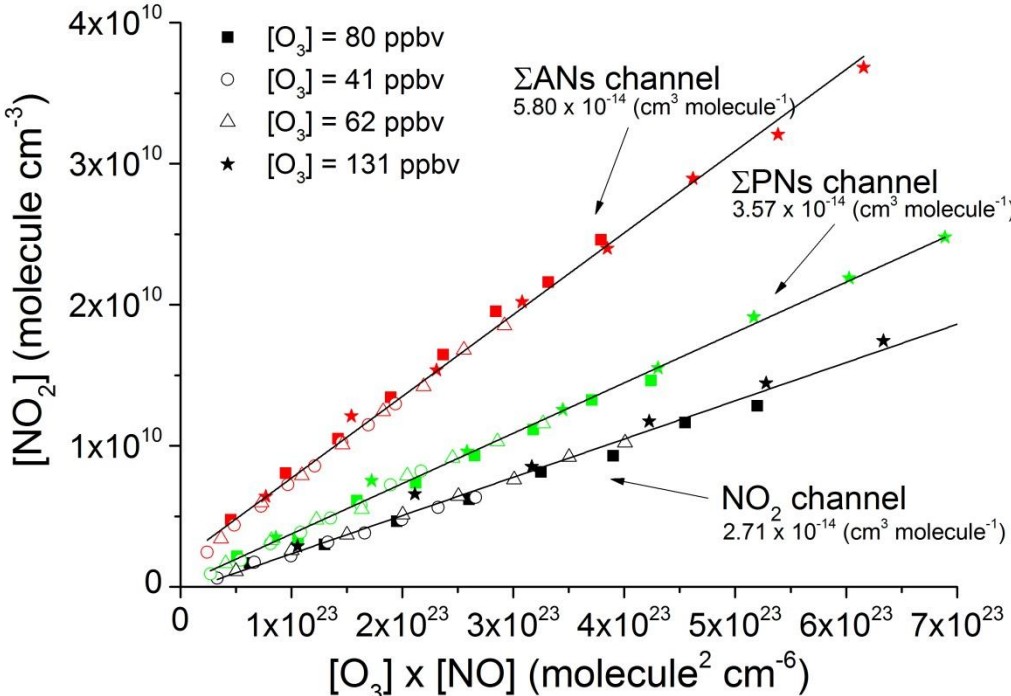

**Figure 9:** Production of NO$_2$ from the O$_3$ + NO reaction when sampling into the three 405 nm channels with different amounts of NO and

5    O$_3$. The numbers in parentheses are the slopes of least-squares fits (solid lines) which correspond to the product of the effective second

order rate constant and the reaction time.





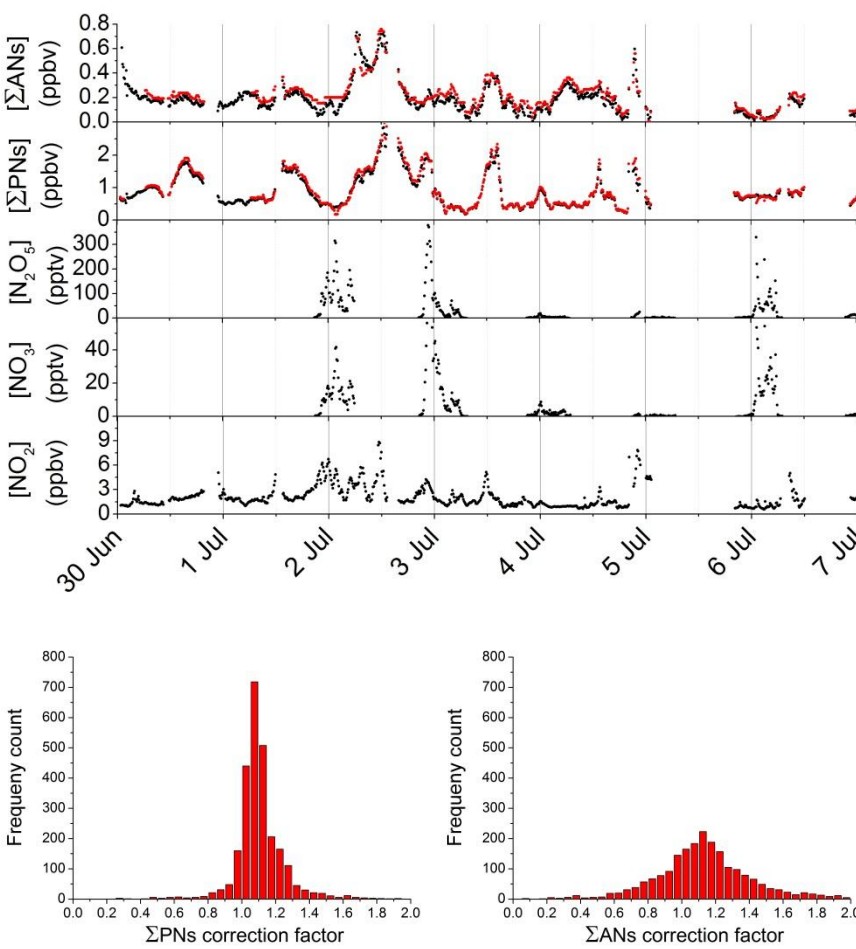

**Figure 10. (a)** Mixing ratios of NO$_2$, NO$_3$, N$_2$O$_5$, ΣPNs and ΣANs measured by the TD-CRDS instrument between the 30[th] June and 7[th] of July 2015 at the "Kleiner Feldberg" Observatory, Germany as part of the NOTOMO campaign. For the three lowest panels in ([NO$_2$], [NO$_3$] and [N$_2$O$_5$]), the black points represent the final corrected data. For the ΣPNs panel, the black points correspond to uncorrected data obtained by subtracting [NO$_2$] measured in the NO$_2$ channel from the NO$_2$ measured in the ΣPNs channel. The red points have been corrected using the chemical model with the iterative fitting procedure. For the ΣANs panel, the black points correspond to the uncorrected data (obtained by subtracting the NO$_2$ signal measured in the ΣPNs channel from the NO$_2$ signal measured in the ΣANs channel). The red points have been corrected using the chemical model plus iterative fitting procedure. Panels **(b)** and **(c)** are frequency distributions for the correction factors applied using the iterative numerical simulations.





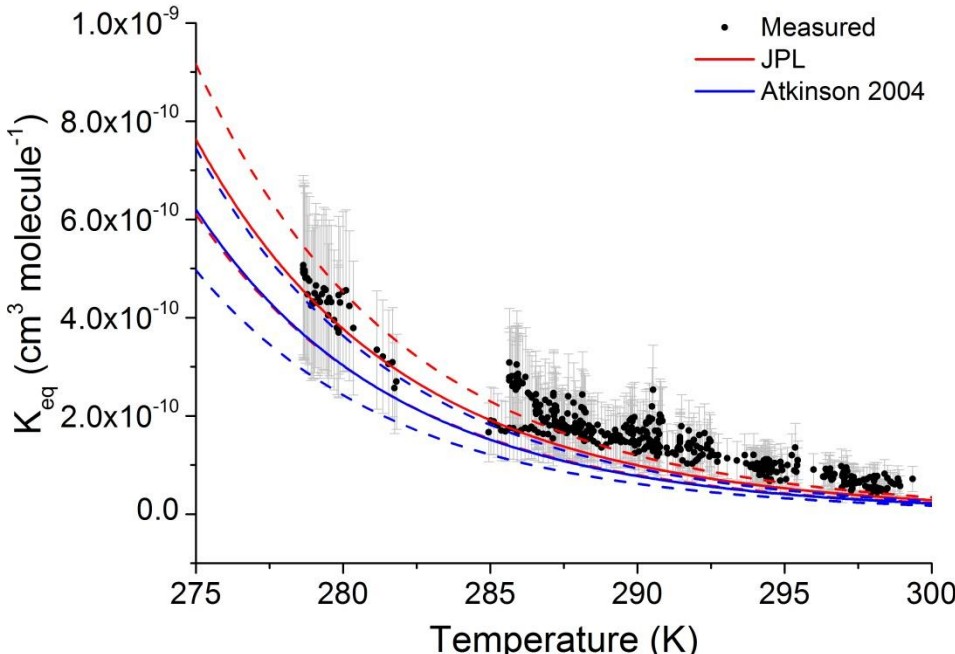

**Figure 11:** Equilibrium constant ($K_{eq}$) for the reaction $NO_2 + NO_3 = N_2O_5$ calculated from measurement of each trace gas (black points, with grey bars representing total uncertainty). The solid lines are values of $K_{eq}$ recommended by the JPL evaluation panel (red line, (Burkholder et al., 2016)) and IUPAC evaluation committee (blue line, (Atkinson et al., 2004)). The dashed lines represent the minimum and maximum values according to the evaluations.