# Peer review of "A 5-channel cavity ring-down spectrometer for the detection of NO2, NO3, N2O5, total peroxy nitrates and total alkyl nitrates"

_Atmospheric Measurement Techniques, 2016_

## Referee Comment (RC1) · Anonymous Referee #1 · 11 Aug 2016

This manuscript reports the development, characterization, and field deployment of a 5 channel cavity ring-down instrument for measuring several components of oxidized nitrogen. Several design improvements over previous versions of similar instruments are discussed including an addition of a zero-air humidification system. An extensive laboratory characterization of potential interferences due to chemistry occurring within the heated inlets is performed and satisfactorily interpreted using an inlet model. The manuscript is thorough, well organized, and provides details useful for other groups performing similar measurements. I recommend publication following attention to the comments listed below.

General Comments:

On page 6 it is mentioned that the dissociation temperature for the PNs heated inlet during ambient measurements was set to 15 K above the plateau temperature from lab (line 17) and similar steps were taken for the ANs inlet. During times of stability in the concentrations of NO2, PNs, and ANs it is possible to conduct these types of temperature scans in ambient air. Were these types of field scans performed during the ambient measurements to verify the use of this temperature during the field deployment?

Although small alkyl nitrates such i-propyl nitrate used in the laboratory experiments have negligible sampling losses, multifunctional nitrates such as those formed from biogenic volatile organic compounds are more "sticky." Has transmission of these types of ANs through the filter or through the tubing prior to the heated section been evaluated? How does this impact the measurements of ANs during the NOTOMO campaign where one would expect to be influenced by multifunctional nitrates derived from biogenics?

Although a lack of HNO3 decomposition in the ANs channel is clear, did the authors increase the temperature further to see at what point HNO3 decomposition started? This would be useful both in the context of the lab measurements and also the field data.

Specific Comments:

Pg 1 line 21: should read "all reactive oxidized nitrogen species"

Pg 2 line 23-24: Individual nitrates can also be measured via chemical ionization mass spectrometry (e.g., Beaver et al., 2012). Given the thoroughness of the discussion, it seems prudent to include this.

Pg 10 line 6: Please explain what parts of the heated inlet A, B, C, and D refer to in the main text and not just the supplement.

Figure 4: The wording "black circles" in the caption referring to HNO3 could be confusing since solid black circles (dots) are also used. I recommend changing the wording to "open black circles." Supplement pg 2 last line: Please correct urge to purge.

Supplement Fig. 1: Please convert temperature to K to be consistent with the main text.

Reference Beaver, M. R., Clair, J. M. S., Paulot, F., Spencer, K. M., Crounse, J. D., LaFranchi, B. W., Min, K. E., Pusede, S. E., Wooldridge, P. J., Schade, G. W., Park, C., Cohen, R. C. and Wennberg, P. O.: Importance of biogenic precursors to the budget of organic nitrates: observations of multifunctional organic nitrates by CIMS and TD-LIF during BEARPEX 2009, Atmos Chem Phys, 12, 5773–5785, doi:10.5194/acp-12-5773-2012, 2012.

---

## Referee Comment (RC2) · Anonymous Referee #2 · 27 Aug 2016

General comments:

Sobanski et al. describe a 5 channel instrument based on cavity ring down spectroscopy for measurement of NO2, peroxy nitrates, alkyl nitrates, NO3 and N2O5. The first three are measured directly as NO2 or by thermal conversion to NO2, where NO2 is measured using CRDS at 405 nm. The latter two are measured directly as NO3 or by thermal conversion to NO3 using 662 nm CRDS. Although all measurements have been described previously by this group, this paper summarizes the performance characteristics of an instrument that detects all 5 simultaneously. It also adds to and augments the thorough description from this group of the radical chemistry and wall loss corrections required for measurement of this set of 5 reactive trace gases. I recommend publication following attention to the specific comments below.

Specific comments:

Page 2, lines 23-25: Reference should be made to CIMS techniques developed more recently that can detect speciated organic nitrates.

Page 5, line 19: "ms" presumably means microseconds, not milliseconds?

Page 5, line 22: does "zero signal" mean with continuous NO added or under a flow of zero air?

Page 5, line 31: Comment on the potential for thermal dissociation of N2O5 or PAN at the elevated 305 K temperature in the NO2 channel.

Page 6, line 2: Why is the optical isolator unnecessary at 405 nm? Empirically determined, or is there a clear reason for it?

Page 7, line 27: Suggest replacing the phrase "essentially calibration free" with "absolute measure of concentration within the optical cavity" or equivalent phrase. Calibration free implies no requirement for standard additions, which is never the case in practice, even for non-reactive trace gases measured using optical instruments.

Page 8, line 4: Is the uncertainty in the NO3 transmission through the filter and housing really as low as 3%, even for sampling ambient air with aerosol accumulation on the filter? Some further comment here is warranted.

Page 11, bottom: How significant is the reaction sequence leading to, for example, alpha lactone production? Perhaps this is discussed further in Thieser 2016, but there is no referencing given in this paragraph to justify what appears to be a somewhat arbitrary sequence of radical reactions.

Section 3.2.4. Two comments. First (minor), the approximation k16[O2] » k17[NO2] should be noted with respect to equation (3). Second (more important), is this treatment realistic for ambient air, in which there may be reactions of atomic O with other

species that reduce the effect of R17? The authors should comment.

Page 13, lines 27-28: Confusing sentence structure.

Section 3.2.5: How is NO + O3 affected by thermal dissociation of O3 referenced above? Presumably this reduces the influence of the NO + O3 reaction directly, but then requires accounting of O + NO -> NO2? Please comment.

Page 16, lines 15-16: The derived equilibrium constant agrees to within the combined uncertainty, but the field determination is systematically larger. Give the average deviation of this difference and note that the field data do not scatter around the denter line of either recommendation.

Page 16, line 11: Remove the characterization of the correction as "rather small" (arbitrary here, a subset of values exceed 50%) but instead give only the center value and the width of the distribution, which is visually symmetric enough that a Gaussian fit may be appropriate.
* * *

---

## Author Comment (AC2) · 30 Aug 2016

We thank the referee for the positive evaluation of our manuscript. Our replies to each comment (in black) are listed below. Red text indicates changes to the manuscript.

| Referee 2 |
|---|
| |
| Sobanski et al. describe a 5 channel instrument based on cavity ring down spectroscopy for measurement of NO2, peroxy nitrates, alkyl nitrates, NO3 and N2O5. The first three are measured directly as NO2 or by thermal conversion to NO2, where $NO_2$ is measured using CRDS at 405 nm. The latter two are measured directly as $NO_3$ or by thermal conversion to NO3 using 662 nm CRDS. Although all measurements have been described previously by this group, this paper summarizes the performance characteristics of an instrument that detects all 5 simultaneously. It also adds to and augments the thorough description from this group of the radical chemistry and wall loss corrections required for measurement of this set of 5 reactive trace gases. I recommend publication following attention to the specific comments below |
| *Specific Comments* |
| Page 2, lines 23-25: Reference should be made to CIMS techniques developed more recently that can detect speciated organic nitrates. |
| We added a reference to the work by Beaver et al. (Beaver et al.,2012) |
| Page 5, line 19: "ms" presumably means microseconds, not milliseconds |
| ms in this case means millisecond, it means the full decay acquired is 1500 μs which correspond to 10 times the ring-down time (≈150 μs). The first 1200 μs only are used to extract the ring-down time which allows a calculation of its value with a high signal-to-noise ratio. |
| Page 5, line 22: does "zero signal" mean with continuous NO added or under a flow of zero air? |
| In this case, the signal was corresponding to zero air but the same results are obtained with a flow of $NO_3$ containing air mixed with a high amount of NO. The comment "(obtained in this experiment by sampling zero air)" is added to the text. |
| Page 5, line 31: Comment on the potential for thermal dissociation of N2O5 or PAN at the elevated 305 K temperature in the NO2 channel. |
| According to IUPAC recommandations, the rate constant for the thermal decomposition of PAN at 303 K and atmospheric pressure is $3.3 \times 10^{-4}$ /s which implies that less than 1‰ of initial PAN is decompose in 1.5 s.  For $N_2O_5$ the decomposition rate constant is $4.4 \times 10^{-2}$ /s which implies a decomposition efficiency of ≈ 6 %. The highest $N_2O_5$ to $NO_2$ ratio measured during the NOTOMO campaign was 17% which results in a maximum contribution of ≈ 1% $NO_2$ from $N_2O_5$ decomposition. We add this value in the total $NO_2$ error. Text added in Sect. 3.2.7. |
| Page 6, line 2: Why is the optical isolator unnecessary at 405 nm? Empirically determined, or is there a clear reason for it? |
| It was found empirically that back reflections were either negligible or if not, did not have an influence on the 405 nm LD emission spectrum. A comment is added to the text. |
| Page 7, line 27: Suggest replacing the phrase "essentially calibration free" with "absolute measure of concentration within the optical cavity" or equivalent phrase. Calibration free implies no requirement for standard additions, which is never the case |

in practice, even for non-reactive trace gases measured using optical instruments.

"CRDS method is essentially calibration free" replaced by "CRDS is an absolute concentration measurement technique"

Page 8, line 4: Is the uncertainty in the NO3 transmission through the filter and housing really as low as 3%, even for sampling ambient air with aerosol accumulation on the filter? Some further comment here is warranted.

The 3 % uncertainty was obtained by repeated measurements under laboratory conditions. During the NOTOMO campaign, no discontinuities in the $NO_3$ signal were observed after hourly filter changes, which implies that there was no measureable change in transmission over the hour of exposure. In highly polluted environments, or those with highly reactive aerosol this may not be the case and more frequent filter changes may be necessary to avoid loss of $NO_3$. This is now mentioned in the text.

Page 11, bottom: How significant is the reaction sequence leading to, for example, alpha lactone production? Perhaps this is discussed further in Thieser 2016, but there is no referencing given in this paragraph to justify what appears to be a somewhat arbitrary sequence of radical reactions.

The reaction sequence is not arbitrary, but is based on experimental and theoretical kinetic studies. These studies have identified α-lactone as the main product of CH2C(O)OOH thermal decomposition (Carr et al., 2011) and also as a significant product of the reaction between CH3CO and $O_2$ (Tyndall et al., 1995; Carr et al., 2007, 2011; Chen and Lee, 2010; Groß et al., 2014; Papadimitriou et al., 2015). These references were cited in Thieser et al.,2016 and are now repeated in the revised manuscript.

Section 3.2.4. Two comments. First (minor), the approximation k16[O2] » k17[NO2] should be noted with respect to equation (3). Second (more important), is this treatment realistic for ambient air, in which there may be reactions of atomic O with other species that reduce the effect of R17? The authors should comment

First comment: details on the steady state O atom concentration calculation added to the text.
Second comment: The loss rate of $NO_2$ depends on the O atom steady state concentration. The dominant loss term for O atoms is $O + O_2$ with a pseudo first order loss rate of circa 2000 /s at 650 K and atmospheric pressure. No other process makes a significant contribution to the O atom loss. For example, taking 10 ppbv of isoprene and a maximum O atom rate coefficient of $2 \times 10^{-10}$ $cm^3$/molecule/s, results in a pseudo first order, O-atom loss rate constant of only 20/s.

Page 13, lines 27-28: Confusing sentence structure

Original sentence replaced by : "For the ΣPNs channel (448 K), using this expression results in an underestimation of the effective (measured) production of $NO_2$ by a factor 1.06. For the ΣANs (648 K) channel the equivalent factor is 1.52"

Section 3.2.5: How is NO + O3 affected by thermal dissociation of O3 referenced above? Presumably this reduces the influence of the NO + O3 reaction directly, but then requires accounting of O + NO -> NO2? Please comment.

The change in ozone concentration by thermal decomposition is taken into account in the effective $NO + O_3$ rate constant derived from the laboratory experiment (see Fig 9).
The second part of the comment refers to the fact that NO is generated from $O + NO_2$. The amount of NO generated is however under normal conditions too small (100 to 200 ppt) to introduce a significant bias to the correction.

Page 16, lines 15-16: The derived equilibrium constant agrees to within the combined uncertainty, but the field determination is systematically larger. Give the average deviation of this difference and note that the field data do not scatter around the center

line of either recommendation.

We want to avoid over interpretation of the deviation between laboratory and field derived equilibrium constants and prefer not to be quantitative. We have changed the text and now indicate possible causes for the deviation observed. "Within combined uncertainty the values agree, though we note that the JPL parameterisation results in values that are closer to those obtained by analysing the field measurements and that the agreement is better at lower temperatures. We are wary of over interpretation of this fact and aware that the laboratory determinations that led to the recommended values are expected to be more accurate, especially close to room temperature. In this context we note that small errors in the measurement of the temperature or a 10-20 % inlet loss of $NO_3$ would have been difficult to observe but would have a significant impact on the equilibrium constant calculated from the field data and may have contributed to the differences observed."

Page 16, line 11: Remove the characterization of the correction as "rather small" (arbitrary here, a subset of values exceed 50%) but instead give only the center value and the width of the distribution, which is visually symmetric enough that a Gaussian fit may be appropriate.

Sigma value for a gaussian fit are added to the text. Sentence : "The peak of the distributions are at $\approx 1.1$ indicating firstly that the corrections required are dominated by $NO_2$ rather than NO, and secondly that the corrections needed are rather small" replaced by "The peak of the distributions are at $\approx 1.1$ for both datasets and the sigma values corresponding to a Gaussian fit are respectively 0.06 and 0.18 for $\Sigma$PNs and $\Sigma$ANs indicating that the corrections required are dominated by $NO_2$ rather than NO, and that the corrections for the $\Sigma$ANs are varying more due to the presence of more organic radicals in the $\Sigma$ANs channel"

---

## Author Comment (AC1)

We thank the referee for the positive evaluation of our manuscript. Our replies to each specific comment (in black) are listed below. Red text indicates changes to the manuscript.

| Referee 1 |
|---|
| |
| This manuscript reports the development, characterization, and field deployment of a 5 channel cavity ring-down instrument for measuring several components of oxidized nitrogen. Several design improvements over previous versions of similar instruments are discussed including an addition of a zero-air humidification system. An extensive laboratory characterization of potential interferences due to chemistry occurring within the heated inlets is performed and satisfactorily interpreted using an inlet model. The manuscript is thorough, well organized, and provides details useful for other groups performing similar measurements. I recommend publication following attention to the comments listed below |
| *General Comments* |
| On page 6 it is mentioned that the dissociation temperature for the PNs heated inlet during ambient measurements was set to 15 K above the plateau temperature from lab (line 17) and similar steps were taken for the ANs inlet. During times of stability in the concentrations of NO2, PNs, and ANs it is possible to conduct these types of temperature scans in ambient air. Were these types of field scans performed during the ambient measurements to verify the use of this temperature during the field deployment? |
| This type of experiment was not conducted during the NOTOMO campaign but it is planned for the next field deployment of this and a similar instrument that we are developing for NO$y$ measurement. |
| Although small alkyl nitrates such i-propyl nitrate used in the laboratory experiments have negligible sampling losses, multifunctional nitrates such as those formed from biogenic volatile organic compounds are more "sticky." Has transmission of these types of ANs through the filter or through the tubing prior to the heated section been evaluated? How does this impact the measurements of ANs during the NOTOMO campaign where one would expect to be influenced by multifunctional nitrates derived from biogenics? |
| At present we have no indication that ANs are lost in the inlet. Our sampling strategy during NOTOMO (high volume flow rates through a wide inlet) should reduce wall losses in the inlet to a minimum. |
| Although a lack of HNO3 decomposition in the ANs channel is clear, did the authors increase the temperature further to see at what point HNO3 decomposition started? This would be useful both in the context of the lab measurements and also the field data. |
| We could not go to higher temperatures as the heating system was not set-up for this. A future instrument to measure NO$y$ (including HNO$_3$) will be able to examine this. |
| *Specific Comments* |
| Pg 1 line 21: should read "all reactive oxidized nitrogen species" |
| "reactive nitrogen species" replaced by "reactive oxidized nitrogen species" |
| Pg 2 line 23-24: Individual nitrates can also be measured via chemical ionization mass spectrometry (e.g., Beaver et al., 2012). Given the thoroughness of the discussion, it seems prudent to include this. |
| Reference added to the text and text added : "or Chemical Ionization Mass Spectrometry (CIMS) (Beaver et al., 2012)" |
| Pg 10 line 6: Please explain what parts of the heated inlet A, B, C, and D refer to in the main text and not just the supplement. |

Changes applied as asked by the referee. Added text : "[…] section D of the reactor (part of the cylindrical reactor between the fritted glass and the cavity inlet, see Fig. S1) [...]" and "[…] in section B (containing the glass beads) rises […]"

Figure 4: The wording "black circles" in the caption referring to HNO3 could be confusing since solid black circles (dots) are also used. I recommend changing the wording to "open black circles."

"black circles" replaced by "open black circles"

Supplement pg 2 last line: Please correct urge to purge.

"urge" replaced by "purge"

Supplement Fig. 1: Please convert temperature to K to be consistent with the main text.

Figure changed accordingly.